# Ribonucleic Acid Sequencing Reveals the Upregulation and Resolution of Inflammation and Extracellular Matrix Remodeling in Lidocaine-Treated Human Acute Monocytic Leukemia Cell Line THP-1

**DOI:** 10.3390/biomedicines12030509

**Published:** 2024-02-23

**Authors:** Sheng-Wei Feng, Wei-Chun Lin, I-Ta Lee, Sheng-Dean Luo, Ching-Shuen Wang

**Affiliations:** 1School of Dentistry, College of Oral Medicine, Taipei Medical University, Taipei 110, Taiwan; shengwei@tmu.edu.tw (S.-W.F.); itlee0128@tmu.edu.tw (I.-T.L.); 2Division of Prosthodontics, Department of Dentistry, Taipei Medical University Hospital, Taipei 110, Taiwan; 3School of Dental Technology, College of Oral Medicine, Taipei Medical University, Taipei 110, Taiwan; weichun1253@tmu.edu.tw; 4Department of Otolaryngology, Kaohsiung Chang Gung Memorial Hospital, Chang Gung University College of Medicine, Kaohsiung 833, Taiwan; rsd0323@cgmh.org.tw

**Keywords:** local anesthetic, anti-inflammation, macrophage polarization, resolution of inflammation, wound healing

## Abstract

Lidocaine, a local anesthetic widely used in dentistry, is esteemed for its efficacy and safety. Recent research reveals its additional role in modulating the immune system, and particularly in reducing inflammation crucial for protecting tooth-supporting tissues. Notably, monocytes and macrophages, essential cellular components overseeing various physiological and pathological processes, stand as potential mediators of lidocaine’s effects. Therefore, this study aimed to investigate how lidocaine influences cell behavior using RNA sequencing. To investigate the effect of lidocaine on THP-1 cells’ behavior, we performed an MTT assay and RNA-Seq along with qPCR analyses to evaluate the transcriptomic and proteomic changes in THP-1 cells. Our results showed that a high dose of lidocaine (>1 mM) had a significant cytotoxic effect on THP-1 cells. However, a lidocaine dose lower than 0.5 mM induced a mixed anti-inflammatory profile by significantly upregulating tissue remodeling (*GDF15*, *FGF7*, *HGF*, *COL4A3*, *COL8A2*, *LAMB2*, *LAMC2*, *PDGFRA*, and *VEGFA*) and through the resolution of inflammation (*Cpeb4*, *Socs1*, *Socs2*, *Socs3*, *Dusp1*, *Tnfaip3*, and *Gata3*) gene cassettes. This study explores the effect of lidocaine on the THP-1 in the M2-like healing phenotype and provides potential applications of lidocaine’s therapeutic effectiveness in dental tissue repair.

## 1. Introduction

Achieving profound local anesthesia is pivotal for successful dentoalveolar extraction. A fundamental principle in local anesthesia is to ensure deep numbness with minimal medication and tissue penetration, thereby reducing the risk of complications and patient anxiety [1,2]. These anesthetic agents function by reversibly binding to sodium channels, blocking sodium’s entry into cells and consequently impeding nerve impulse propagation [3]. This prevents nociceptive impulses from reaching the brain, effectively eliminating pain perception. Local anesthesia refers to the loss of sensation in a specific body area due to nerve ending depression or peripheral nerve conduction inhibition. Lidocaine, the first amide-linked anesthesia agent, synthesized in 1943, has undergone extensive evaluation for its efficacy and safety [1].

When necessitating tooth extraction, dentists commonly administer local anesthetics to enhance patient comfort during the procedure while ensuring procedural ease [2,4]. After tooth removal, the incurred wound initiates an inflammation process, pivotal in combating foreign pathogens and advancing wound healing [5,6]. However, uncontrolled inflammation may hinder the healing process if it becomes excessive [7]. This condition could potentially advance to cellulitis if unaddressed [8]. Past research underscores the significance of macrophage engagement in wound healing. Its pivotal function encompasses inflammation mitigation, cellular debris clearance, and the expediting of wound repair [9,10,11].

Macrophages, immune cells traced back to monocyte transformation via signal stimulation [12], showcase remarkable versatility in numerous physiological roles such as antigen–antibody uptake, phagocytosis, wound healing, bone resorption, antimicrobial defense, and antigen presentation [12]. They can be polarized into two major subtypes: pro-inflammatory M1 and healing M2. M1 macrophages drive pathogen phagocytosis and provoke inflammation, while M2 macrophages foster anti-inflammatory actions and tissue repair. Further subdivision leads to M2a (wound healing), M2b (immune regulation, tumorigenesis), M2c (immune suppression), and M2d (tumor angiogenesis) types [10]. Positioned as early responders at injury sites, macrophages critically regulate inflammatory processes. If modulating macrophage polarization, guiding pro-inflammatory macrophages towards the M2 subtype for repair becomes feasible; it could control diseases by reducing excess inflammation and facilitate tissue regeneration.

Lidocaine, initially developed as a local anesthetic and antiarrhythmic agent, distinguishes itself from its precursor, cocaine, by omitting the hallucinogenic and addictive components [13]. Originally employed for cardiac arrhythmias, lidocaine’s analgesic potential emerged later [1]. Recent pharmacological and clinical advances have uncovered its immunoregulatory capabilities, notably inhibiting anti-inflammatory responses and mitigating acute lung injury [14,15]. A noteworthy sensitization to anticancer drugs has also been documented [16,17]. Contemporary research underscores lidocaine’s efficacy in suppressing inflammation within lipopolysaccharide-activated macrophages [18]. As previously mentioned, macrophage phenotypes can be classified into four subtypes, and there is currently no research investigating which specific subtype of M2 macrophages lidocaine treatment leads to. This study aimed to discover the phenotypic transition and transcriptomic changes in macrophages upon lidocaine stimulation. This investigation’s outcomes stand to unravel the molecular dynamics of lidocaine’s involvement in tissue repair, explore the possibility of repurposing an existing drug, and expedite the drug development process.

## 2. Materials and Methods

### 2.1. THP-1 Cell Culture

Cells from the human monocytic cell line THP-1 (BCRC, Hsinchu, Taiwan) were cultured in RPMI 1640 medium with 2 mM L-glutamine adjusted to contain 1.5 g/L sodium bicarbonate, 4.5 g/L glucose, 10 mM HEPES, 1.0 mM sodium pyruvate, 90% fetal bovine serum, 10% supplemented with 0.05 mM 2-mercaptoethanol (Sigma-Aldrich, St. Louis, MO, USA). Cells with a density of 1 × 10^6^ cells/mL were stimulated using 25 nM of phorbol-12-myristate-13-acetate (Sigma-Aldrich) for 48 h and another 24 h of resting before lidocaine treatments.

### 2.2. Cell Viability

THP-1 cells were seeded at 150,000 cells/well in 24-well plates and cultured as described earlier. Various concentrations of lidocaine (from 0 mM (untreated) to 10 mM (lidocaine-treated)) were used to stimulate cells at 0, 6, 12, and 24 h. After incubation with lidocaine, the supernatant was removed, and cells were rinsed prior to adding MTT (Sigma-Aldrich) to develop color. The media were then removed and 1 mL of lysis buffer (SDS 30%/N,N-dimethyl-formamide at ratio 2:1, pH 4.7) was added to each well. Plates were then incubated at 37 °C and gently agitated at 70 rpm for 1 h. The absorbance was then measured at 570 nm. For microscopic observations, the THP-1 cells at various treatment conditions were imaged with a Nikon E200 LED microscope (Nikon, Tokyo, Japan).

### 2.3. RNA Sequencing

THP-1 cells treated with 0 and 0.5 mM lidocaine for 12 h were analyzed with RNA-sequencing. Total RNA was isolated from lidocaine induced and non-induced control cells using TRIzol reagent (Thermo Fisher, Waltham, MA, USA). RNA purity was checked using the NanoPhotometer^®^ spectrophotometer (IMPLEN, München, Germany) and RNA integrity was assessed using the RNA Nano 6000 Assay Kit of the Bioanalyzer 2100 system (Agilent Technologies, Santa Clara, CA, USA). Then, RNA samples with three replicates were delivered to a company (Genomics, New Taipei City, Taiwan) for RNA-sequencing on an Illumina Hiseq 2500 platform and 150 bp paired-end reads were generated.

### 2.4. Bioinformatic Analysis

SeqPrep was used to perform quality control for the sequences after RNA-seq. After adapters were removed, sequences with lengths below 25 bp were discarded, followed by the trimming of low-quality bases and the deletion of sequences with N ratios higher than 10%. The read depth, error rate (%), Q20 and Q30 values, and GC-content (%) of the resulting high-quality clean reads were then evaluated. DEG (Differentially Expressed Gene) analysis was performed using DESeq2 and edgeR [19,20], and DEGs with log2FC > 1 and *p*-adjust < 0.05 were considered to be significantly differently expressed genes. The Database for Annotation, Visualization, and Integrated Discovery (DAVID) was used to systematically extract biological information from numerous genes [21,22] and to perform GO enrichment and Kyoto Encyclopedia of Genes and Genomes (KEGG) pathway analysis. *p* < 0.05 in the analysis was considered to indicate a statistically significant difference.

### 2.5. Reverse Transcription Polymerase Chain Reaction

Following treatment with 0 and 0.5 mM lidocaine for 12 h, the RNA stabilization solution was removed on the next day, and total RNA was extracted using the RNAeasy Kit (QIAGEN, Hilden, Germany) according to the manufacturer’s instructions. One µg of total RNA was reverse-transcribed into 20 µL cDNAs with the iScript Reverse Transcription Kit according to the manufacturer’s instructions (Bio-Rad, Hercules, CA, USA). The reactions were carried out by adding the following reagents: 1 μM of each primer (stock was prepared at 10 μM; Appendix A), 25 ng cDNA, and 10 μL of 2 × SYBR green master mixes (Bio-Rad) in a total of 25 μL. The Polymer Chain Reaction was performed on 96-well plates at the following temperature cycles: Step 1: 95 °C for 5 min; Step 2: 95 °C for 30 s, 60 °C for 30 s, and 72 °C for 35 s for 35 more cycles; and Step 3: 72 °C for 5 min. Relative fold changes of gene expression were normalized using β-actin, and the results were plotted and analyzed using Prism software 8.3 (GraphPad Software, Boston, MA, USA).

### 2.6. Statistical Analysis

Three independent experiments with cell lines from different donors were performed for each test. Data are shown as mean ± standard error of the mean (SEM). A post hoc test was employed following ANOVA to assess differences among the groups using Prism (GraphPad Software, San Diego, CA, USA). A *p*-value of less than 0.05 was considered statistically significant.

## 3. Results

### 3.1. High Concentrations of Lidocaine Affect Macrophage Viability

The MTT method was used to determine cell viability after treatment with lidocaine for 0–24 h (Figure 1A). The viability of the macrophages was not statistically affected by lidocaine at lower concentrations, ranging from 0.1 to 0.5 mM at 6 h and 12 h (Figure 1A). However, macrophage viability appeared to be reduced slightly after a longer exposure to lidocaine at 12 h, and was significantly affected at 24 h exposure at concentrations higher than 1 mM (Figure 1A). Moreover, we observed that lidocaine has a time-dependent effect on the viability of macrophages, meaning that longer exposures caused more damage to macrophages. Bright-field microscopic observations are consistent with MTT results. Representative images of various treatments are shown in Figure 1B, which showed concentration- and time-dependent effects of lidocaine on cell morphology. Lidocaine at lower concentrations (0.1 and 0.5 mM) was found to not alter morphology (Figure 1B). On the other hand, higher concentrations of lidocaine (i.e., 1, 2, 5 and 10 mM) were seen to significantly alter cellular morphology at 12 and 24 h. We concluded that elevated concentrations of lidocaine exhibited a time-dependent impact. This effect was observed to hinder cell proliferation and manifest the existence of clusters of deceased cells (indicated by arrows in Figure 1B).

To further verify the cytotoxic effects of lidocaine, we next examined the gene expressions of cell proliferation markers *Ki67* and *Myc* at the mRNA levels in THP-1 cells. As shown in Figure 1C, there was a significant reduction in the mRNA expression of *Ki67* and *Myc* in macrophages at concentrations higher than 0.5 mM (~0.01%) at 24 h. In contrast, lidocaine at concentrations lower than 0.5 mM did not affect cell proliferation with shorter exposure times to lidocaine at 6 and 12 h (Figure 1C).

### 3.2. Transcriptome Analysis of Lidocaine-Treated Macrophages Revealed Significant Upregulation of Tissue Remodeling Cassettes in Macrophages

To verify the underlying mechanisms behind the lidocaine on macrophages, RNA-seq was conducted to determine the possible genes and signaling pathways that could be involved in this process. Samples from two groups were used for comparison: untreated only (Un) and lidocaine-treated (Lido). After sequencing, differentially expressed gene (DEG) analysis was performed to identify gene expression changes among these two groups. After comparing the genes in the lidocaine-treated and non-treated macrophages, a scatter plot showed that there were 2282 DEGs in lidocaine-treated macrophages compared to untreated, including 1012 upregulated genes and 1270 downregulated ones (Figure 2A). To describe the functions of the DEGs obtained from RNA-Seq more generally, we analyzed the obtained genes with Database for Annotation, Visualization, and Integrated Discovery (DAVID) to obtain the gene ontology (GO) terms of the DEGs. GO term assignment and enrichment analyses showed that multiple GO terms were significantly enriched in lidocaine-treated groups (e.g., signaling transduction, cell adhesion, cell differentiation of the plasma membrane, and integral components of extracellular membrane). GO terms with q < 0.05 were identified as significantly enriched. According to the functional enrichment and gene ontology results, several terms were significantly enriched within each category (CC, BP, and MF) (Table 1).

Moreover, KEGG pathway enrichment was performed to explore the enriched pathways for DEGs. As compared to the untreated group, 16 KEGG pathways were identified as significantly enriched in the lidocaine-treated macrophages (Table 2). It is worth noting that the most enriched pathways in both upregulated and downregulated groups were neuroactive ligand–receptor interactions and pathways in cancer (Table 2).

Additionally, the specific pathways of PI3K-Akt, cGMP-PKG, Oxytocin, and MAPK signaling pathways were also identified in upregulated DEGs (Table 2). In contrast, only the MAPK signaling pathway was identified in downregulated DEGs (Table 2).

Since there is no KEGG category regarding macrophage polarization, therefore, the analysis of DEGs related to macrophage polarization was conducted manually. Consistent with previous findings [15,23], specific DEGs selected in untreated and lidocaine-treated groups showed that lidocaine reduces inflammatory responses by decreasing the gene expression of Tumor Necrosis Factor Alpha (*Tnfa*), Interleukin 1 Beta (*Il1b*), Cluster of Differentiation 38 (*Cd38*), Cluster of Differentiation 80 (*Cd80*), G Protein-Coupled Receptor 18 (*Gpr18*), Toll-Like Receptor 4 (*Tlr4*), and Prostaglandin-Endoperoxide Synthase 2 (*Ptgs2*) (Figure 2B). In contrast, lidocaine treatment increases the anti-inflammatory gene expression of Interferon Regulatory Factor 6 (*Irf6*), Kinase Insert Domain Receptor (*Kdr*), Programmed Cell Death 1 Ligand 2 (*Pcdc1lg2*), Breast Cancer Anti-Estrogen Resistance 3 (*bcar3*), Matrix Metalloproteinase 12 (*mmp12*), Von Willebrand Factor (*vwf*), Suppressor of Cytokine Signaling 2 (*socs2*), Integrin Subunit Beta 3 (*Itgb3*), Interleukin 10 (*Il10*), Purinergic Receptor P2Y1 (*P2ry1*), and Aquaporin 9 (*Aqp9*) (Figure 2C). Intriguingly, among these DEGs, we found that genes associated with wound healing (i.e., growth factor secretion, ECM remodeling and angiogenesis) were upregulated in lidocaine-treated macrophages (Figure 2D). Specifically, Growth Differentiation Factor 15 (*Gdf15*), Fibroblast Growth Factor 7 (*Fgf7*), Hepatocyte Growth Factor (*Hgf*), Collagen Type IV Alpha 3 Chain (*Col4a3*), Collagen Type VIII Alpha 2 Chain (*Col8a2*), Laminin Subunit Beta 2 (*Lamb2*), Laminin Subunit Gamma 2 (*Lamc2*), Platelet-Derived Growth Factor Receptor Alpha (*Pdgfra*), and Vascular Endothelial Growth Factor A (*Vegfa*) were upregulated in lidocaine-treated DEGs as compared to the untreated group (Figure 2D). Upon further analysis, utilizing functional protein association networks (STRING), it was revealed that these genes are correlated with extracellular matrix organization and remodeling (see Appendix A).

### 3.3. Lidocaine Increases the Resolution of Inflammation-Associated Gene Expression in Macrophages

Macrophages exhibit a remarkable degree of phenotypic plasticity, enabling them to adeptly react to a wide array of stimuli [12,24]. This plasticity empowers macrophages to orchestrate a spectrum of functions aimed at finely regulating inflammation, the resolution of inflammation, and tissue repair mechanisms. Therefore, we sought to understand whether lidocaine has an impact on modulating the plasticity of macrophages. Among these DEGs, a panel of macrophage polarization markers of M2 subtypes (M2a–d) were used to determine the effects under lidocaine treatment. Our results showed that 0.5 mM lidocaine polarizes resting THP-1 into a mixed M2 subtype by significantly increasing the gene expression levels of Mrc1, Fabp4, Plin2, Dsc1, Vegfa, and Il-10 (Figure 3A) in these DEGs.

To further assess the influence of lidocaine on the resolution of the inflammation pathway, an in-depth characterization of lidocaine-treated DEGs using multiple markers associated with the resolution of inflammation pathway was undertaken. Notably, existing research in the scientific literature suggests that the G-protein-coupled receptor FPR2 plays a pivotal role in modulating anti-inflammatory responses and initiating the resolution of inflammation, which is integral to the maintenance of tissue homeostasis [25,26,27]. Furthermore, essential enzymes and proteins, including alox5, alox15, and alox5ap, have been identified as critical components of the pro-resolving process [28,29]. The perturbation of these enzymatic entities has demonstrated a capacity to alter macrophage polarization towards an M1 phenotype that did not favor wound healing. Our results demonstrated that lidocaine significantly decreased the inflammatory markers of *Fpr2*, *alox5, and Il1rl1* gene expression in THP-1. In contrast, the resolution of inflammation markers such as *Cpeb4*, *Socs1*, *Socs2*, *Socs3*, *Dusp1*, *Tnfaip3*, *Gata3*, and *IRG1* was significantly increased under lidocaine treatment (Figure 3A).

Further validation of these RNA-Seq results was performed by quantitative real-time PCR (qPCR) using six genes, *Fpr2*, *Alox5*, *Il1rl1*, *Cpeb4*, *Socs1*, and *Gata3* (Appendix A). We screened different concentrations of lidocaine at three time points on THP-1, and found similar results for RNA-Seq. Specifically, we found that pro-inflammatory markers *FPR2*, *ALOX5*, and *IL1RL1* were downregulated with 0.5 mM lidocaine treatment at both 6 and 12 h (Figure 3B). In contrast, the resolution of inflammation markers *CPEB4*, *SOCS1*, and *GATA3* was upregulated with 0.5 mM lidocaine treatment at both 6 and 12 h (Figure 3B). At 24 h treatment, we did not observe significant changes in these tested genes. Together with our findings on the anti-inflammatory function of lidocaine (e.g., the upregulation of *CPEB4*, *SOCS1*, and *GATA3*), we concluded that lidocaine promotes the resolution of inflammatory responses in resting THP-1 cells.

## 4. Discussion

Apart from its established role as an analgesic in dental clinical practice [1,30], lidocaine has recently been revealed to possess immunomodulatory capacities. These encompass the inhibition of neutrophil migration and accumulation, decreases in macrophage phagocytosis, and the activation of cytotoxic functions within killer cells. Particularly noteworthy is its involvement in regulating the anti-inflammatory abilities of macrophages, an area that has begun to be explored. Our experimental findings are in concurrence with earlier research, as we have found that small doses of lidocaine (<0.5 mM, ~0.01%) can lower the overall inflammatory response of macrophages. Our KEGG enrichment pathway analysis has also unveiled an elevation in neuropeptide oxytocin signaling among lidocaine-treated macrophages. Notably, lidocaine’s anti-inflammatory influence becomes apparent at concentrations lower than those necessary for sodium channel blockades [31]. It is imperative to note that lidocaine’s effect on inflammation, particularly concerning inflammatory polymorphonuclear granulocytes (PMNs), macrophages, and monocytes, is independent of sodium channel blockade [15]. Notably, our findings revealed that macrophages display reduced tolerance to lidocaine in comparison to other cell types like fibroblasts [32]. Specifically, even at a 1 mM concentration (0.02%) of lidocaine, macrophages demonstrate toxicity. This concentration is significantly lower than the 4.27 mM lidocaine (~0.1%) toxicity reported for fibroblasts in prior investigations [33]. This fourfold contrast in tolerance underscores the imperative to thoroughly consider dosage-dependent immunotoxicity when utilizing lidocaine within clinical scenarios.

By modulating excessive inflammatory responses in macrophages, numerous studies have established a connection between this modulation and accelerated tissue repair. A prior clinical study [34] indicated that lidocaine effectively expedites wound healing after melanoma resection. The authors also observed a significant increase in CD31+ cell count following lidocaine treatment. They speculated that lidocaine might influence the proliferation or aggregation of CD31+ cells, subsequently impacting tissue repair, given the close association of CD31/PECAM-1 with angiogenesis and tissue repair. Interestingly, CD31 is not solely expressed in endothelial cells; it has also been identified in macrophages in substantial quantities [34]. This suggests that under the influence of lidocaine, CD31+ macrophages could potentially contribute to an accelerated tissue repair process, thus justifying the need for future investigations in this direction.

Along with our findings, this has suggested that lidocaine might affect the behavior of macrophages towards a healing subtype (Figure 2 and Figure 3). For instance, our results (Figure 2C,D) revealed the upregulation of critical factors associated with the wound healing process (such as *IL-10*, *AQP9*, *RUNX2*, *TGFβ*, *COL8A2*, *GDF15*, and others) due to lidocaine exposure [35,36]. Furthermore, it has been demonstrated that the elevated expression of RUNX2 is pivotal for the differentiation of mesenchymal stem cells into osteoprogenitor cells, particularly immediately after tooth extraction [6,37].

Nevertheless, the shortage of studies has not demonstrated that lidocaine orchestrates macrophage phenotype modulation via CD31 to facilitate tissue repair. Furthermore, the intricate molecular mechanism behind lidocaine’s effects lacks in vivo validation as yet. Despite the absence of a precise molecular understanding, gel-based products containing lidocaine, such as Astero^TM^, are already accessible in the market for addressing pain-associated diabetic wounds, supported by clinical trial findings that attest to their significant efficacy in wound repair [38]. Our experimental outcomes provide supplementary validation by confirming that lidocaine unequivocally exerts a direct influence on macrophage phenotypes. Based on the observations in Figure 3, we observed a polarization shift of the macrophage phenotype toward the M2 subtype upon exposure to lidocaine. While it is regrettable that this study cannot definitively discern which of the M2a–d subtypes lidocaine shifts the macrophages towards, an overall examination of transcriptome expression indicates that lidocaine does transform THP-1 cells into a M2-like macrophage subtype. Moreover, it increases the expression of growth factors, elevates molecules related to TGFβ signaling, enhances macrophage ECM remodeling capability, and augments angiogenesis potential (Figure 2D). In general, an increase in M2 macrophages is associated with pain relief and closely linked to wound repair [11,39]. Suppressing the inflammatory response of M1 macrophages accelerates the resolution of tissue inflammation, consequently expediting tissue repair [40,41].

## 5. Conclusions

In conclusion, this study has explored the impact of lidocaine on macrophages, revealing that even at low doses, lidocaine can mitigate macrophage inflammatory responses while concurrently enhancing the expression of genes associated with tissue repair and inflammation resolution. While the focus was primarily on the transcriptomic changes in lidocaine-induced THP-1 cell behavior, this investigation sheds light on the potential of lidocaine in treating inflammatory diseases related to macrophages. Moreover, there is considerable potential for refining lidocaine’s formulation and developing targeted therapeutic strategies to address macrophage-related disorders.

## 6. Limitations

Future research should prioritize protein and functional assays and relevant animal models to further elucidate inflammation resolution and extracellular matrix remodeling mechanisms under lidocaine treatment.

## Figures and Tables

**Figure 1 biomedicines-12-00509-f001:**
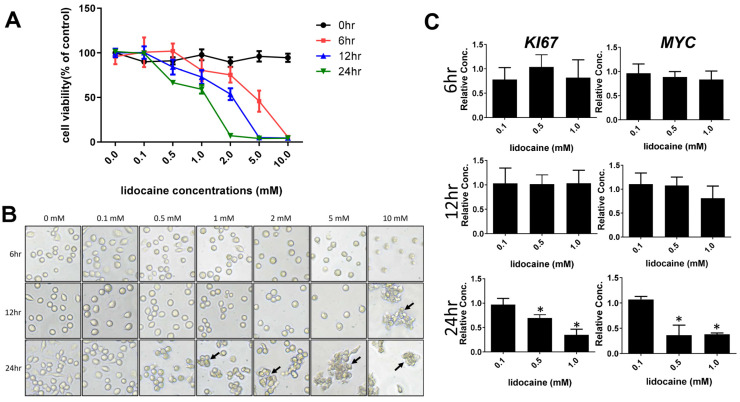
Cell proliferation measurement by MTT colorimetric assay and RT-qPCR. (**A**) Viability of THP-1 cells exposed to various lidocaine concentrations for 0, 6, 12, and 24 h; (**B**) cell morphology visualized using light microscope (20× magnification) at various lidocaine concentrations for 0, 6, 12, and 24 h, where arrows indicate significant morphological changes in THP-1 cells; and (**C**) qPCR analysis of KI67 and MYC markers in THP-1 treated with or without lidocaine for 6, 12, and 24 h. The results from three independent experiments are presented as the mean ± SD. * *p* < 0.05.

**Figure 2 biomedicines-12-00509-f002:**
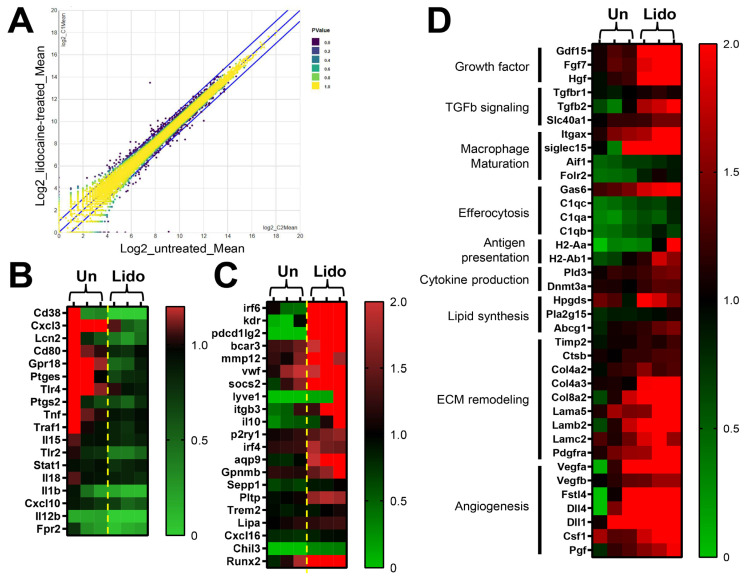
Identification of DEGs among untreated and 0.5 mM lidocaine-treated THP-1 samples for 12 h. (**A**) The scatter plot shows the DEGs in lidocaine-treated compared to untreated samples. (**B**) The heat map shows the DEGs related to pro-inflammatory genes in lidocaine-treated compared to untreated samples. (**C**) The heat map shows the DEGs related to anti-inflammatory genes in lidocaine-treated compared to untreated samples. (**D**) The heat map shows the DEGs related to ECM assembly genes in lidocaine-treated compared to untreated samples.

**Figure 3 biomedicines-12-00509-f003:**
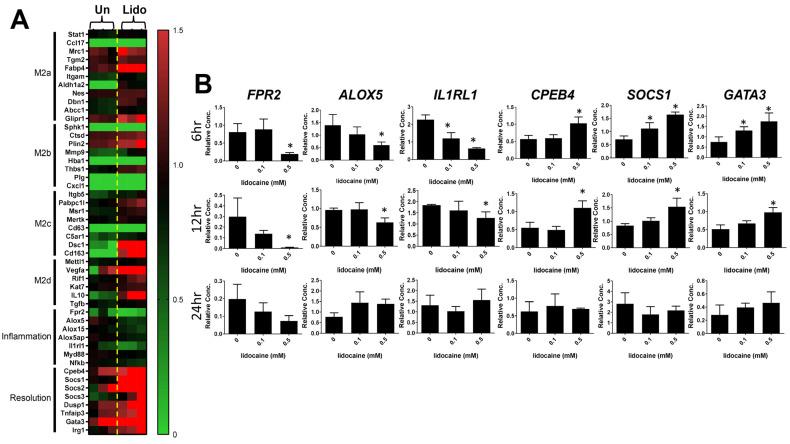
Identification of DEGs along with qPCR validation among untreated and lidocaine-treated THP-1 samples. (**A**) The heat map shows the DEGs related to M2a, M2b, M2c, M2d, Inflammatory, and resolution of inflammation genes in 0.5 mM lidocaine-treated compared to untreated samples. (**B**) qPCR analysis of FPR2, ALOX5, IL1RL1, CPEB4, SOCS1, and GATA3 markers in THP-1 treated with or without lidocaine (0, 0.1 and 0.5 mM) for 6, 12, and 24 h (* *p*  <  0.05, as compared to the control). The results from three independent experiments are presented as the mean ± SD. * *p* < 0.05.

**Table 1 biomedicines-12-00509-t001:** The top 20 GO terms of BP, CC, and MF of upregulated and downregulated genes. Abbreviations: BP, biological process; CC, cellular component; MF, molecular function.

**(A) The top 20 enriched GO terms of upregulated genes**			
Category	Term	Count	*p* Value
BP	signal transduction	64	1.9 × 10^−5^
BP	cell adhesion	50	4.3 × 10^−12^
BP	cell differentiation	40	4.4 × 10^−5^
BP	proteolysis	26	5.2 × 10^−4^
BP	nervous system development	23	4.7 × 10^−3^
BP	protein phosphorylation	23	5.9 × 10^−2^
BP	spermatogenesis	22	2.1 × 10^−2^
CC	plasma membrane	224	4.7 × 10^−11^
CC	integral component of membrane	210	2.2 × 10^−7^
CC	extracellular region	90	3.4 × 10^−4^
CC	integral component of plasma membrane	89	6.9 × 10^−12^
CC	extracellular exosome	78	4.0 × 10^−2^
CC	extracellular space	71	3.2 × 10^−2^
CC	cell surface	37	1.4 × 10^−4^
CC	apical plasma membrane	34	3.7 × 10^−8^
CC	synapse	34	3.3 × 10^−5^
CC	neuron projection	29	2.7 × 10^−6^
CC	external side of plasma membrane	29	2.7 × 10^−4^
CC	dendrite	24	5.1 × 10^−3^
CC	glutamatergic synapse	22	7.5 × 10^−3^
MF	identical protein binding	61	7.5 × 10^−2^
MF	calcium ion binding	47	1.9 × 10^−6^
MF	protein homodimerization activity	38	1.0 × 10^−3^
MF	sequence-specific double-stranded DNA binding	30	2.2 × 10^−3^
MF	transcription factor activity, sequence-specific DNA binding	29	4.6 × 10^−3^
MF	protein serine/threonine/tyrosine kinase activity	24	5.5 × 10^−3^
MF	receptor binding	22	9.0 × 10^−3^
**(B) The top 20 enriched GO terms of downregulated genes**			
Category	Term	Count	*p* Value
BP	signal transduction	83	2.6 × 10^−4^
BP	cell adhesion	57	6.1 × 10^−9^
BP	nervous system development	50	2.5 × 10^−10^
BP	cell differentiation	44	1.1 × 10^−2^
BP	chemical synaptic transmission	36	1.6 × 10^−9^
BP	positive regulation of gene expression	35	9.0 × 10^−3^
BP	cell–cell signaling	34	1.4 × 10^−9^
BP	positive regulation of cell proliferation	33	4.6 × 10^−2^
BP	proteolysis	30	1.0 × 10^−2^
BP	inflammatory response	30	1.2 × 10^−2^
BP	intracellular signal transduction	29	4.2 × 10^−2^
BP	brain development	27	7.2 × 10^−5^
BP	axon guidance	26	7.5 × 10^−7^
BP	positive regulation of cell migration	26	1.9 × 10^−4^
BP	cell surface receptor signaling pathway	26	3.8 × 10^−3^
BP	homophilic cell adhesion via plasma membrane adhesion molecules	24	1.9 × 10^−6^
BP	positive regulation of protein phosphorylation	22	3.9 × 10^−4^
BP	extracellular matrix organization	20	4.7 × 10^−4^
BP	visual perception	20	2.2 × 10^−3^
CC	plasma membrane	346	3.9 × 10^−19^
CC	integral component of membrane	309	2.5 × 10^−9^
CC	extracellular region	157	5.5 × 10^−11^
CC	extracellular space	138	6.0 × 10^−9^
CC	integral component of plasma membrane	136	1.0 × 10^−18^
CC	Golgi apparatus	63	3.2 × 10^−2^
CC	synapse	58	2.6 × 10^−10^
CC	cell surface	53	1.4 × 10^−5^
CC	dendrite	52	2.0 × 10^−10^
CC	glutamatergic synapse	49	2.9 × 10^−10^
CC	neuronal cell body	46	2.2 × 10^−10^
CC	neuron projection	43	7.5 × 10^−9^
CC	axon	41	3.6 × 10^−8^
CC	perinuclear region of cytoplasm	40	9.8 × 10^−2^
CC	Golgi membrane	38	6.2 × 10^−2^
CC	endoplasmic reticulum lumen	36	1.4 × 10^−7^
CC	apical plasma membrane	33	4.1 × 10^−4^
CC	extracellular matrix	30	2.1 × 10^−6^
CC	external side of plasma membrane	26	3.7 × 10^−2^
CC	perikaryon	26	2.3 × 10^−8^
MF	calcium ion binding	72	2.1 × 10^−9^
MF	receptor binding	36	1.9 × 10^−4^
MF	sequence-specific double-stranded DNA binding	36	3.0 × 10^−2^
MF	macromolecular complex binding	28	5.9 × 10^−3^
MF	growth factor activity	22	1.9 × 10^−5^
MF	cytokine activity	22	1.5 × 10^−4^
MF	integrin binding	21	2.8 × 10^−5^
MF	transmembrane signaling receptor activity	21	3.1 × 10^−4^
MF	signaling receptor activity	21	2.9 × 10^−3^

**Table 2 biomedicines-12-00509-t002:** KEGG pathway enrichment analyses of DEGs in unregulated and downregulated groups.

**(A) The KEGG pathway enrichment analysis of upregulated genes**		
Term	Count	*p* Value
Pathways in cancer	30	9.4 × 10^−4^
Neuroactive ligand–receptor interaction	25	2.1 × 10^−4^
Calcium signaling pathway	22	1.7 × 10^−5^
PI3K-Akt signaling pathway	19	1.7 × 10^−2^
Cytokine–cytokine receptor interaction	16	3.0 × 10^−2^
MAPK signaling pathway	15	6.4 × 10^−2^
cGMP-PKG signaling pathway	14	1.3 × 10^−3^
Hematopoietic cell lineage	13	3.2 × 10^−5^
Platelet activation	13	2.9 × 10^−4^
Oxytocin signaling pathway	13	2.0 × 10^−3^
**(B) The KEGG pathway enrichment analysis of downregulated genes**		
Term	Count	*p* Value
Neuroactive ligand–receptor interaction	37	2.3 × 10^−6^
Pathways in cancer	32	4.6 × 10^−2^
Cytokine–cytokine receptor interaction	24	4.2 × 10^−3^
Calcium signaling pathway	21	5.9 × 10^−3^
MAPK signaling pathway	21	3.4 × 10^−2^

## Data Availability

The data presented in this study are available on request from the corresponding author.

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
