# Peer review of "Ribonucleic Acid Sequencing Reveals the Upregulation and Resolution of Inflammation and Extracellular Matrix Remodeling in Lidocaine-Treated Human Acute Monocytic Leukemia Cell Line THP-1"

_biomedicines, 2024, doi:10.3390/biomedicines12030509_

Round 1

Reviewer 1 Report

Comments and Suggestions for Authors

The authors aiming to discover the phenotypic transition and transcriptomic changes in macrophages upon lidocaine stimulation. This investigation's outcomes stand to unravel the molecular dynamics of lidocaine's involvement in tissue repair, explore the possibility of repurposing an existing drug, and expedite the drug development process. Major point about their finding is that they have not shown any direct interaction between these molecules (GDF15, FGF7, HGF, COL4A3, COL8A2, LAMB2, LAMC2, PDGFRA, and VEGFA). It would be useful if authors could investigate any direct or indirect interaction that supports their proposed model in Figure 2. There are some minor points which can be addressed by the authors:
•       Tense form of the language in methods needs correction throughout the section.
•       Abbreviations used in manuscript could be described at least once in entire manuscript. (for example: line 193 GDF15)

•       Grammatical errors can be checked throughout the manuscripts.
•       Figures can be labeled better to improve readability (For example, the font of   Figure 1-3 are blurred)
•       In the figures for “X and Y” axis labels are missing (For example, Figure 2A)
•       The manuscript also needs English proofreading.

Comments on the Quality of English Language

The manuscript also needs English proofreading.

Author Response

Dear Reviewers:

We thank the editor and reviewers for their thorough review of our proposed manuscript, and their valuable comments. We have carefully addressed your comments. Here, we have detailed the changes made to the manuscript and answered the reviewers' comments. We also re-wrote the part with the most repetition as possible as we could.

Thank you again for your valuable time.

Reviewer 2 Report

Comments and Suggestions for Authors

Thank you for your submission. The article, titled " RNA Sequencing reveals the upregulation of resolution of inflammation and ECM remodeling in lidocaine-treated human THP-1 cell line," is a research paper, and the aim of this study was to investigate how lidocaine affects THP-1 cell behavior through RNA sequencing. The focus of this paper is primarily on lidocaine-induced transcriptomic changes in THP-1 cell behavior, but this study reveals the potential of lidocaine in the treatment of inflammatory diseases associated with macrophages, as well as providing potential applications for the therapeutic role of lidocaine in oral tissue repair. However, several minor limitations need to be revised before publication:

1. The relationship between lidocaine and dentistry is mentioned in the Abstract and Discussion sections, but not in the Introduction section. And the whole text is only a very brief description of the relationship between dentistry and lidocaine, which is not very relational. It is suggested that the relationship between lidocaine and dentistry should be reorganized in the Introduction section.

2. What does it mean that CC, BP and MF, which appear in the text on line 173, do not have full names?

3. A comparative analysis of different bioactive materials, including their advantages, limitations, and efficacy in various contexts, would provide a clearer understanding of their relative merits.

4. There are many errors in the text: show in line 197 should be changed to shows. THP-1 cells, multiple misspellings in lines 150, 153, 196, and 241. In addition, please keep the units uniform throughout the text. Is the unit mM/ml correct?

5. Repetition of Fig. 1A, Fig. 1B , Fig. 1C in the main text, is it sufficient for these to appear only once?

6. The text below the bars in the image should be kept at a uniform height on the left and right sides.

7.Most recent studies such as Biomaterials Translational, Viewpoint, 2023 (doi: 10.12336/biomatertransl.2023.02.002), Biomaterials Translational, Review, 2023 (doi: 10.12336/biomatertransl.2023.03.003), Biomaterials Translational, Review, 2023 (doi: 10.12336/biomatertransl.2023.03.004), are recommended to be cited in proper places.

Comments on the Quality of English Language

There is a lot of room for improvement in response to the quality of the English language in this paper. There are a number of errors and mistakes in the text such as spelling, singular and plural, etc., which need to be reconfirmed for the textual content.

Author Response

Dear Reviewers:

We thank the editor and reviewers for their thorough review of our proposed manuscript and valuable comments. We have carefully addressed the reviewers’ comments. Here, we have detailed the changes made to the manuscript and answered comments from the reviewers. We also re-wrote the part with the most repetition as possible as we could.

Thank you again for your valuable time.

Reviewer 3 Report

Comments and Suggestions for Authors

The authors have investigated how lidocaine influences cell behavior using RNA sequencing.

effect of lidocaine on the THP-1 cells were also monitored. The q-PCR was performed for several genes  (tissue remodeling genes (GDF15, FGF7, HGF, COL4A3, COL8A2, LAMB2, LAMC2, PDGFRA, and VEGFA) and inflammation genes (Cpeb4, Socs1, Socs2, Socs3, Dusp1, Tnfaip3, and Gata3)) and their upregulation and down regulation was reported. I have the following comments:

Technical comments:

While doing cell culture of THP-1 cells, the authors have used 2-mercaptoethanol and also for induction have used phorbol-12-myristate-13-acetate. During MTT assay, do these chemicals react with the MTT? Please explain the chemical reactions of MTT and these two compounds and how do they affect the MTT results.

In the discussion, the dose of lidocaine is given as 0.02 % (line #259: Specifically, even at a concentration of approximately 0.02% lidocaine, macrophages demonstrate toxicity.), but the cell viability graph (Fig 1 A), shows the dose in mM. It is confusing, please rectify.

The cells were 100 % killed at a dose of 2 mM, after 24 h. This shows that lidocaine is highly toxic to the macrophages. Did you see its toxicity in any other cell lines? How do you justify this result?

The English grammar needs to be corrected for the following sentence:

Line # 61:“This study aiming to discover the phenotypic transition and transcriptomic changes in macrophages upon lidocaine stimulation.”

I recommend minor revision.

Author Response

(The authors gave the same response as above.)
